# Cross-Laminated Timber Floor: Analysis of the Acoustic Properties and Radiation Efficiency

Nicola Granzotto , Arianna Marzi * and Andrea Gasparella

Faculty of Science and Technology, Free University of Bozen, 39100 Bolzano, Italy;
nicolagranzotto74@gmail.com (N.G.); andrea.gasparella@unibz.it (A.G.)
* Correspondence: arianna.marzi@unibz.it

**Abstract:** Cross-Laminated Timber (CLT) is a building technology that is becoming increasingly popular due to its sustainable and eco-friendly nature, as well as its availability. Nevertheless, CLT presents some challenges, especially in terms of impact noise and airborne sound insulation. For this reason, many studies focus on the vibro-acoustic behavior of CLT building elements, to understand their performance, advantages and limitations. In this paper, a 200 mm CLT floor has been characterized in the laboratory, according to ISO standards, by three noise sources: dodecahedron, standard tapping machine and rubber ball. In order to understand the vibro-acoustic behavior of the CLT floor, measurements through the analysis of sound pressure levels and velocity levels, measured by dedicated sensors, were performed. Analysis was carried out in order to understand what is prescribed by the prediction methods available in the literature and by the simulation software. Then, a specific prediction law for the CLT floor under investigation was derived. Finally, an analysis on sound radiation index is provided to complete the vibro-acoustic study.

**Keywords:** CLT; acoustic characterization; sound radiation efficiency

## 1. Introduction

Cross-Laminated Timber (CLT) has increasingly become a timber product of global interest [1]. On the one hand, this is due to (i) the many merits of eco-sustainable wooden building elements, characterized by wide availability in nature, (ii) relative ease of handling and (iii) its environmental friendliness and wide range of end uses [2]. On the other hand, CLT is well known for its suitability [3] in the construction sector and its structural behavior [4]. However, due to their lightweight characteristics, CLT structures are characterized by poor acoustic performance [5]. Many studies were developed to determine if timber component properties are well known among users and the scientific community. The results demonstrate that there is no agreement on this topic [6].

For these reasons, there have been several recent experimental studies to determine the acoustic performance of CLT, with particular attention to its impact on noise behavior [7], and also for comfort perspective on users [8].

For example, Hoeller et al. [9] published the results of a comprehensive experimental study on the sound reduction index of CLT wall and floor systems, and also presented a method usable for calculating the flanking sound transmission, based on the vibrations measurement of sound reduction index, related to different types of panel joints. Schoenwald et al. [10] carried out measurements aimed at quantifying the impact noise of CLT floors, the airborne sound transmission and the lateral transmission, through joints between CLT panels, which were used as a starting database to assess the acoustic performance of a large building structure. Homb et al. [11] collected the results of laboratory measurements on the impact insulation of various CLT floor constructions. Other studies on the impact insulation of different CLT floor systems are presented by Zeitler et al. [12], Di Bella et al. [13] and Caniato et al. [14–16]. Barbaresi et al. [17] presented the results of an experimental campaign on flanking sound transmission.

In this perspective, further studies are needed to develop and improve prediction models of CLT floor sound and vibrational behavior [18,19]. In this regard, the sound radiation index is of paramount importance to understand and simulate the behavior of these elements [20,21]. Hence, to design CLT structures with good acoustic insulation, there is a need to characterize the sound radiation of the vibrating elements. Acoustic radiation has generated increasing interest over the past decades and it has been used in acoustic calculation models, for example, ISO 12354-1 [22].

Thus, to obtain a complete vibro-acoustic characterization and, in particular, to fully understand how material properties impact on the acoustic performance of structural elements, and meet the high quality standards required for CLT floors [23], the sound radiation of CLT floor is an issue that needs to be deepened and studied.

Atalla and Nicolas, in 1994 [24], presented an extensive and detailed bibliographic analysis of the prediction models to compute sound radiation in the same decade, while Nelisse et al. [25] proposed a generalized model for the acoustic radiation from baffled and unbaffled homogeneous plates, with arbitrary boundary conditions. A similar approach was also used by Foin et al. [26] and by Hosseinkhani et al. [27,28], to develop a tool to predict the acoustic and structural vibration response of sandwich plates. More recently, Mejdi and Atalla [29] presented a semi-analytical model to investigate, numerically, the vibro-acoustic response of stiffened plates, while Legault et al. [30] analyzed orthogonally ribbed plates by means of a periodic theory. Rhazi and Atalla [31] used statistical energy analysis and the transfer matrix method to estimate the vibro-acoustic response of mechanically excited multilayer structures.

Considering these studies as a reference, analyses on the vibro-acoustic behavior and radiation efficiency of wooden elements have recently been developed [32]. Fortini et al. [33] characterized the vibro-acoustic behavior of a composite structure and presented a comparison between the sound reduction index predictions and measurements in sound transmission suites, according to ISO 10140-2. Wang et al. [34] compared and validated simulated results with measurements of a timber joist floor, then analyzing the importance of the modes of the case study room and the correlation with the behavior of its floor. Conta and Homb [35] performed experimental investigations to identify the modal properties of such a system and to gain understanding of the sound radiation properties under impact excitation, using both experimental modal analysis (EMA) and the integral transform method (ITM). The results highlight the limitations of standard acoustic laboratories and show the importance of using advance measurement methods to acquire reliable data.

In addition to the studies mentioned above, there are also numerous works dealing with the vibro-acoustic characterisation of wooden floors, in particular with the impact noise [36–40]. Nevertheless, some recent studies have shown the need to study this issue in more detail [41,42].

A recent study [43] has shown, through a Round Robin Test, how the use of software and calculation models, currently available for the study of the characteristics of multi-layer systems, are not reliable when wooden structures are used. For this reason, it is of paramount importance not only to know the characteristics of the materials, but also their radiation efficiency.

The study of sound radiation, in relation to the characterization of CLT floors, is still partially incomplete.

For these reasons, this paper presents: (i) An acoustic investigation of a 200 mm thick CLT floor, conducted in the laboratory, according to ISO 10140 series standards [44–48], both using sound pressure and vibration excitation. The systematic use of the rubber ball as exciting source is included, in order to investigate every above-presented parameter with this noise source. Indeed, if some research focused on the analysis of the impact noise produced by the rubber ball for concrete floors [41,42,49–54], only few studies focused on timber floors [55–58]. (ii) A new mass law for CLT elements in CLT. (iii) In addition to the comparison of the three sound sources, a detailed study on experimental approaches and comparison, with the outcomes of two simulation software for the full characterization of

the CLT floors, was provided. (iv) A complete analysis of the radiation efficiency by varying the type of source. (v) Some simulation methods were compared with existing models.

The novelty of the research consists of studying the acoustic behavior of a CLT floor with three different noise sources and deriving equations that best fit the insulation characteristics of this type of floor.

The article is organized as follows. Section 2 summarizes the physical characteristics of the CLT floor, examined the design of the laboratory used for the measurements and the equations used for the measurements and the theoretical analysis. Section 3 shows the results of measurements, relating to airborne and vibrational noise. Section 4 shows a comparison between simulation methods and measurements. Conclusions are drawn in Section 5.

## 2. Materials and Methods

In this study the impact and airborne sound insulations as well as the radiation efficiency are investigated using sound pressure and velocity sensors on a CLT floor in the laboratory of the Free University of Bolzano, accordingly to ISO 10140-5 [48], to minimize flanking transmission. The laboratory consists of massive structure lined with double layer of gypsum board with a cavity filled with rock wool panels. The test frame and the floor of the receiving room are made of concrete. The two doors of the laboratory have a weighted sound reduction index of 42 dB (Figure 1).

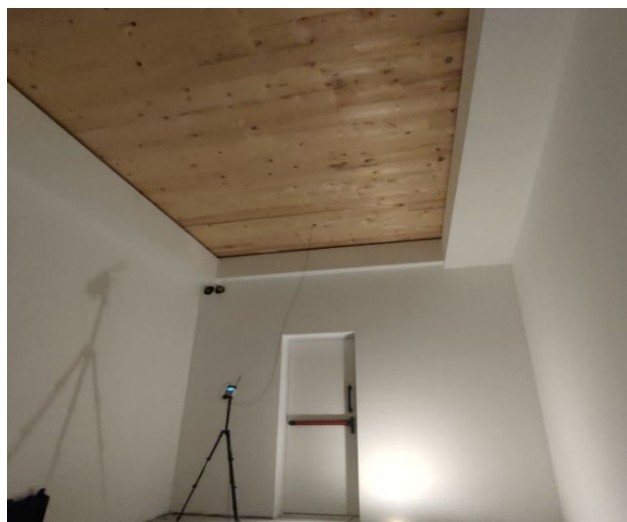

**Figure 1.** Laboratory receiving room and CLT floor.

In order to minimize the variability of the measurements with time a seasoned wood floor was adopted. Thus, for this reason the measurements were all carried out in a short time range (2 consecutive days).

The 5-layer CLT floor has dimensions of 4155 mm (longer length) × 3000 mm (shorter length) with a mass per unit area, $m'$, of 84 kg/m$^2$ and a thickness of 200 mm. The shorter length consists of three resistant timber layers, while the longer length direction consists of two resistant wood layers (Figure 2).

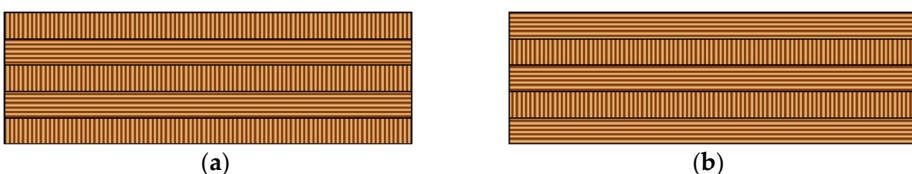

| (a) | (b) |

**Figure 2.** Wood fiber direction. Longer dimension (4155 mm) with 2 resistant layers (**a**). Shorter dimension (3000 m) with 3 resistant layers (**b**).

The CLT floor is made of spruce and it is classified as a C24 according to the EN 338 [59], with the properties described in Table 1.

**Table 1.** Mechanical properties of the CLT under consideration.

| $E_{m,0,mean}$ | $E_{m,90,mean}$ | $\rho_{mean}$ | $m'$ |
|---|---|---|---|
| 11 kN/mm$^2$ | 0.37 kN/mm$^2$ | 420 kg/m$^3$ | 84 kg/m$^2$ |

The CLT floor was laid on an anti-vibration material, 30 mm thick, with a density of 500 kg/m$^3$. The perimeter of the specimen is therefore not bound to the structure.

To characterize the acoustic behavior of the CLT floor's airborne and impact noise, some acoustic indices were determined. With sound pressure levels measurements, the normalized impact sound pressure level with tapping machine, the sound reduction index and the impact sound pressure with rubber ball have been determined. Velocity levels and radiation efficiency levels are then determined combining the vibration and acoustic measurements.

For the measurements, an airborne source and two impact noise sources according to ISO 10140-5 were used (Figure 3). A Svantek 958 four-channel analyzer to acquire sound pressure and vibration data was used.

Dodecahedron Brüel and Kjær Type 4292-L, Power amplifier Brüel and Kjær Type 2734          Tapping machine Brüel and Kjær Type 3207          Rubber ball Norsonic Nor279

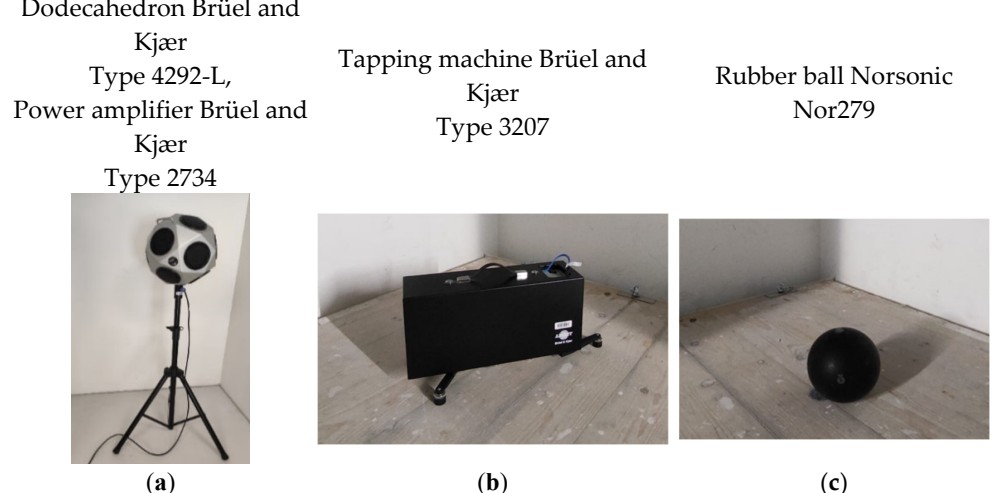

(**a**)                          (**b**)                          (**c**)

**Figure 3.** Airborne noise source (**a**), tapping machine (**b**) and rubber ball (**c**).

An omnidirectional source with twelve loudspeakers (airborne source) driven with a pink noise signal, a standard tapping machine and a rubber ball have been used, respectively. In particular, the rubber ball is useful for a low frequency analysis.

The tapping machine is the one most used both in the study of bare structures and in the case of the application of additional layers [60]. The three sources above described are illustrated in Figure 3.

### 2.1. Acoustic and Vibration Parameters Determined from Measurements

The normalized impact sound pressure level, $L_n$, measured with a standard tapping machine and the energy average maximum impact sound pressure level measured with rubber ball, $L_{i,Fmax}$, are determined according to ISO 10140-3 [46]. Weighted indices, $L_{n,w}$, evaluated from 100 Hz to 3150 Hz, and $L_{iA,Fmax}$, evaluated from 50 Hz to 630 Hz, have been derived according to ISO 717-2 standard [61]. For the measurements with the tapping machine and with the rubber ball, 4 positions of the source (A, B, C, D) and 5 fixed positions of the microphone (R1, R2, R3, R4, R5) have been used as depicted in Figure 4.

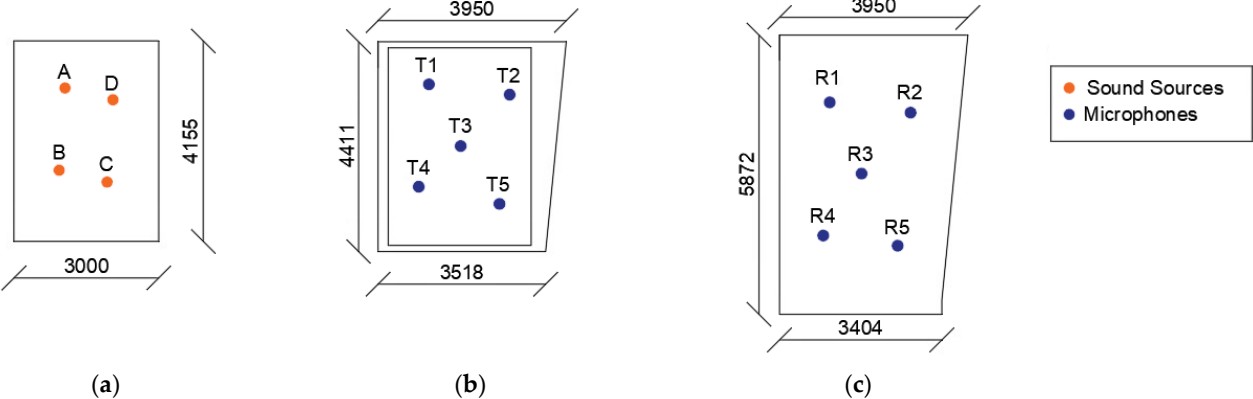

**Figure 4.** CLT floor and sources positions (**a**); microphone positions in the transmitting room (**b**) and in the receiving room (**c**).

The sound reduction index, *R*, is calculated according to ISO 10140-2 [45]. The weighted sound reduction index, $R_{\mathrm{w}}$, is evaluated in the frequency range 100–3150 Hz and is calculated according to ISO 717-1 [62]. For the measurement of the sound reduction index two positions (A, C) of the source and 5 positions of the microphone have been used (Figure 4).

For all the measurements, the microphones have been positioned in a range of heights between 1 and 1.5 metres in order to avoid positions on the same plane.

Velocity levels were measured with an accelerometer fixed in 9 positions with screws. A reference velocity value, $v_0$, of $10^{-9}$ m/s has been considered.

The noise sources (tapping machine, dodecahedron and rubber ball) were positioned at 3 points (A, B, C) for a total of 27 measurements, according to ISO 10848-1 [63] (Figure 5).

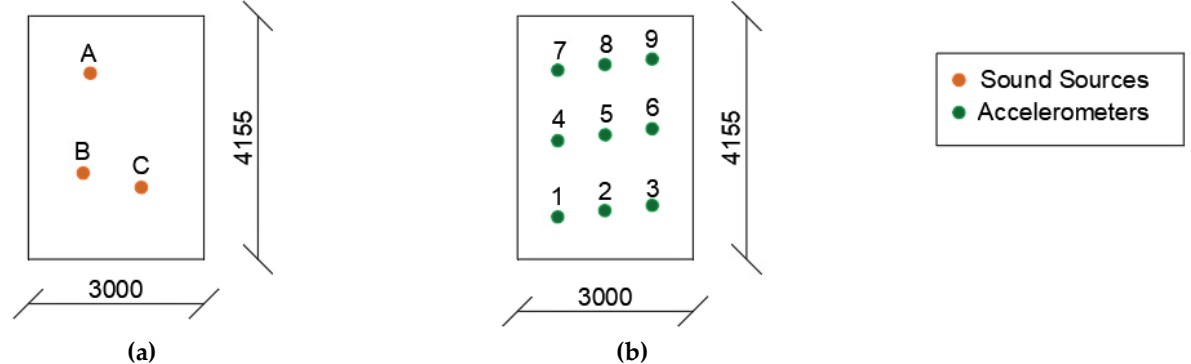

**Figure 5.** General scheme of the positioning of the sound sources (**a**) and the accelerometer (**b**) for the determination of the velocity levels.

Radiation power, $W_{\mathrm{rad}}$, radiation efficiency, $\sigma_{rad}$ and radiation index, $L_{\sigma_{rad}}$, are derived by the Equations (1)–(3) [64]:

$$W_{rad} = A \frac{\langle p^2 \rangle_{S,t}}{4\rho_0 c_0} \left( 1 + \frac{S_T \lambda_0}{8V} \right), \tag{1}$$

$$\sigma_{rad} = \frac{W_{rad}}{S \langle v^2 \rangle_{S,t} \rho_0 c_0} \tag{2}$$

$$L_{\sigma_{rad}} = 10 \lg(\sigma_{rad}) \tag{3}$$

where *A* is the equivalent absorption area of the receiving room, $\langle p^2 \rangle_{S,t}$, is the average sound pressure (with tapping machine, dodecahedron or rubber ball), $\rho_0$ is the air density, $c_0$ is the speed of sound, $S_T$ is the room surface, $\lambda_0$ is the wavelength and *V* is the room

volume, $S$ is the floor surface and $\langle v^2 \rangle_{S,t}$ is the average square vibration velocity (with tapping machine, dodecahedron or rubber ball).

The sound pressure level can be obtained with Equation (4):

$$L_i = L_v + L_{\sigma_{rad}} - 28 + 10 \lg \frac{S}{A} \tag{4}$$

The normalized impact sound pressure level is obtained by the following Equation (5):

$$L_n = L_i + 10 \lg \left( \frac{A}{A_0} \right) \tag{5}$$

The normalized impact sound pressure level in one-third octave bands emitted by the impact of the hammers of the tapping machine is expressed by Equation (6) from ISO 12354-2 [22]:

$$L_n = L_F + 10 \lg(Re(Y)) + 10 \lg(\sigma_{rad}) - 10 \lg(m') + 10 \lg(T_s) + 10.6 \tag{6}$$

where $L_F$ is the force level of the tapping machine (reference $10^{-6}$ N), $Re(Y)$ is the real part of the floor mobility $Y = v/F$, $\sigma_{rad}$ is the radiation efficiency for free bending waves, $m'$ is the mass per unit area, $T_s$ is the structural reverberation time, considered in accordance with the relation expressed by Barbaresi et al. [17] in their study.

For the examined CLT floor the following equation was found which best fit the curve of the measured values:

$$L_{n\_pr} = L_F + 10 \lg(Re(Y)) + 12 \lg(\sigma_{rad}) - 10 \lg(m') + 10 \lg(T_s) + 6.8 \tag{7}$$

As regards the tapping machine, a force value in 1/3 octave band of $\sqrt{0.91f}$ [64] was considered.

In this research, Equation (6) was also used in the case of the rubber ball, considering the maximum force values measured and subtracting the normalization term $10 \lg \left( \frac{A}{A_0} \right)$, as depicted by Equation (8):

$$L_{i,Fmax} = L_{Fmax} + 10 \lg(Re(Y_{max})) + 10 \lg(\sigma_{rad}) - 10 \lg(m') + 10 \lg(T_s) + 10.6 - 10 \lg \left( \frac{A}{A_0} \right) \tag{8}$$

where $L_{Fmax}$ is the maximum force level of the rubber ball, $Re(Y_{max})$ is the real part of the floor mobility.

For the examined CLT floor the following equation was found which best fit the curve of the measured values:

$$L_{i,Fmax\_pr} = L_{Fmax} + 10 \lg(Re(Y_{max})) + 10 \lg(\sigma_{rad}) - 10 \lg(m') + 2 \lg(T_s) + 5 - 10 \lg \left( \frac{A}{A_0} \right) \tag{9}$$

The impact force exposure levels generated by the rubber ball $L_{FE}$ with a reference period of 1 s are reported in Table 2 according to ISO 10140-5 [48].

**Table 2.** Impact force exposure level in each octave band of the rubber ball.

| 1/1 Oct. Band Freq. [Hz] | 31.5 | 63 | 125 | 250 | 500 |
|---|---|---|---|---|---|
| Impact force exposure level [dB re 1 N] | $39.0 \pm 1.0$ | $31.0 \pm 1.5$ | $23.0 \pm 1.5$ | $17.0 \pm 2.0$ | $12.5 \pm 2.0$ |

Since the reported values in Table 2 are not maximum values nor in 1/3 octave bands, the equivalent curve, $L_F$ (1/3 oct) was obtained [13]. To obtain the value correspondent to the reference time of 1 s and then considering the maximum difference in level between

the measured and evaluated ones the conversion values were obtained according to the formula $L_\mathrm{F}$\_max-$L_\mathrm{F}$ (1/3 oct) (Figure 6).

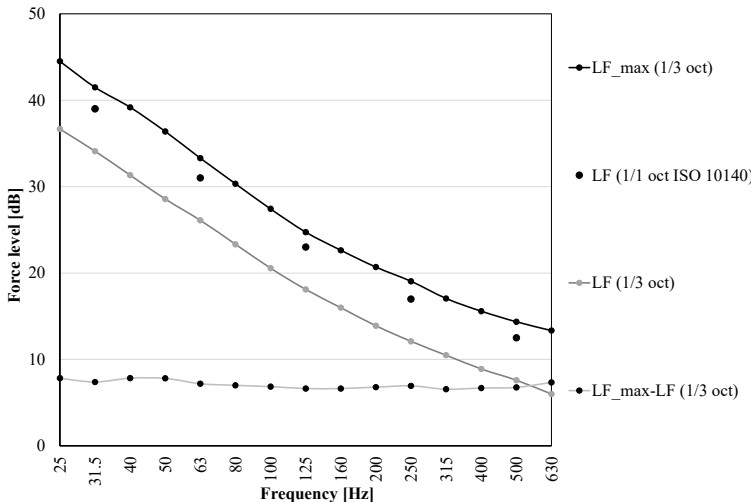

**Figure 6.** Comparison of $L_\mathrm{F}$\_max (1/3 oct), $L_\mathrm{F}$ (1/1 octave), $L_\mathrm{F}$ (1/3 octave), $L_\mathrm{F}$\_max-$L_\mathrm{F}$ (1/3 oct).

### 2.2. Simulations and Analytical Models

Firstly, two commercial software both based on traditional mass law and Sewell correlation [65] were used to compare the simulated versus the measured sound reduction index for the analyzed CLT floor. The elastic properties of the material, such as Young's modulus and the loss factor, were varied, while the real dimensions of the tested slab were considered for the geometry.

Secondly, a numerical model with multi-quadratic interpolation function was used to map the sound radiation index for the three sound sources (tapping machine, dodecahedron, rubber ball). The sound reduction index was determined for the nine positions of the vibration measurement depicted in Section 3.2 permitting the characterization of the average sound pressure level for the three positions of each source.

## 3. Results and Discussion

### 3.1. Experimental Results: Sound Pressure Level-Derived Parameters

The normalized impact sound pressure level obtained using the tapping machine, the sound reduction index retrieved using a dodecahedron and the maximum sound pressure levels measured impinging the rubber ball on the 200 mm CLT floor are shown in Figure 7.

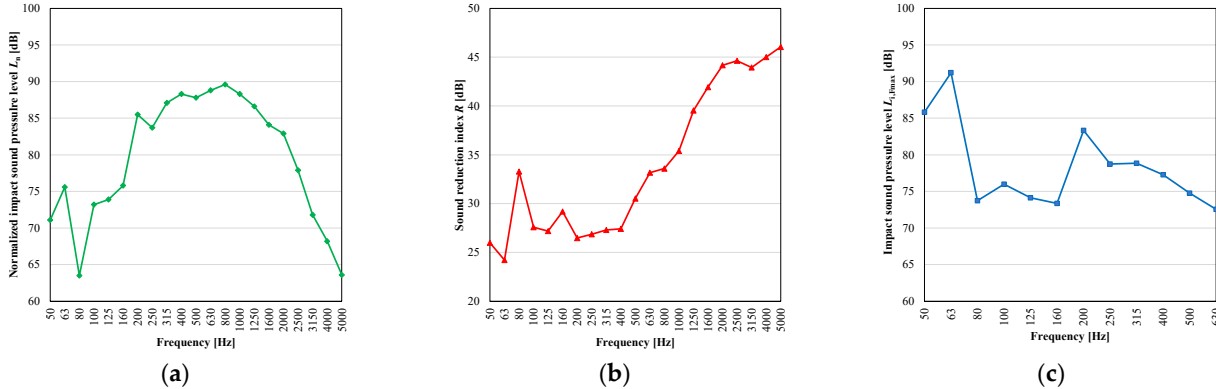

**Figure 7.** Normalized impact sound pressure level (**a**), sound reduction index (**b**), rubber ball impact sound pressure level (**c**) for the 200 mm CLT floor.

For normalized impact sound pressure level, a resonant frequency of 63 Hz is clearly highlighted. Above 800 Hz, the impact of the sound pressure level decreases, as the surface dampens the impact of the hammers by attenuating the impact force. This is due to the fact that the wooden surface of the floor is much less rigid than, for example, a concrete floor, for which this phenomenon is much less noticeable [66] (Figure 8).

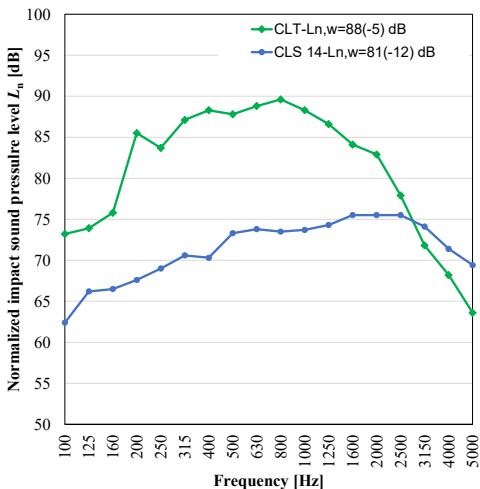

**Figure 8.** Normalized impact sound pressure level for CLT floor (200 mm) and for concrete floor (140 mm).

Due to this important aspect, it is not possible, for example, to use the attenuation data of the impact sound pressure level, obtained on a normalized concrete floor, on wooden floors. The $\Delta L_n$ at high frequency, obtained on the normalized concrete floor of a lining, is greater than the same lining applied on a wooden floor, because floating floors are effective above high frequencies, where the wooden floor has a good impact acoustic behavior, with respect to mass.

When looking at the sound reduction index, *R*, (Figure 7b) a resonant frequency at 63 Hz is also determined. It can be noted that a mass-law-like trend, starting from 315 Hz, is present.

From the comparison of the impact sound pressure level produced by the standard tapping machine (Figure 7c), and the sound reduction index (Figure 7b), it is clear that the peak of *R*, at 80 Hz, corresponds to the minimum impact sound pressure level, as expected [67,68]. Then, in the case of the impact sound pressure level and the rubber ball trends, the resonance frequency is also at 63 Hz.

The weighted index values are depicted in Table 3.

**Table 3.** Weighted normalized impact sound pressure level, $L_{n,w}$, weighted sound reduction index, $R_w$, and A-weighted energy average maximum impact sound pressure level measured with rubber ball according to the standard, $L_{iA,Fmax}$.

| $L_{n,w}$ [dB] | $R_w$ [dB] | $L_{iA,Fmax}$ [dB(A)] |
|:---:|:---:|:---:|
| 88 | 35 | 80 |

The weighted impact sound pressure level and weighted sound reduction index are consistent with the literature [12], confirming the possibility of extending the results obtained in this research to other CLT floors.

In Figure 9, the standard deviations associated to the sound pressure levels and measured for the three sources in the receiving room are compared. Due to the greater variability in the positioning of the source, the rubber ball results are characterized by higher values of standard deviation than the other sources.

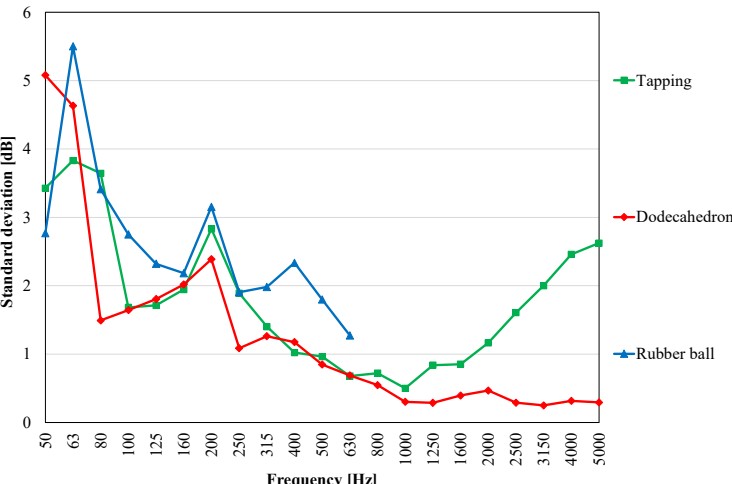

**Figure 9.** Standard deviation of the sound pressure levels of the three sources, measured in the receiving room.

In the case of the impact measurements, it can be noted that due to the non-homogeneous contact surface, the standard deviation is greater than with the airborne measurements. In the literature, the difference is less evident, probably due to the analysis of homogeneous heavy weight floors [69].

Then, by comparing the measured sound reduction index with the ones derived from the mass law $R = 20\lg(mf) - 47$ [70], a new correlation was retrieved experimentally, starting from 315 Hz, according also to the range frequency of the mass law. The proposed mass law for the 200 mm CLT timber floor analyzed here is expressed in Equation (10).

$$R = 17.5\lg(mf) - 50 \tag{10}$$

Then, a comparison of the here-proposed mass law versus the traditional one [70] and the measured sound reduction index is proposed to focus on its reliability. From the results depicted in Figure 10, the following highlights can be drawn:

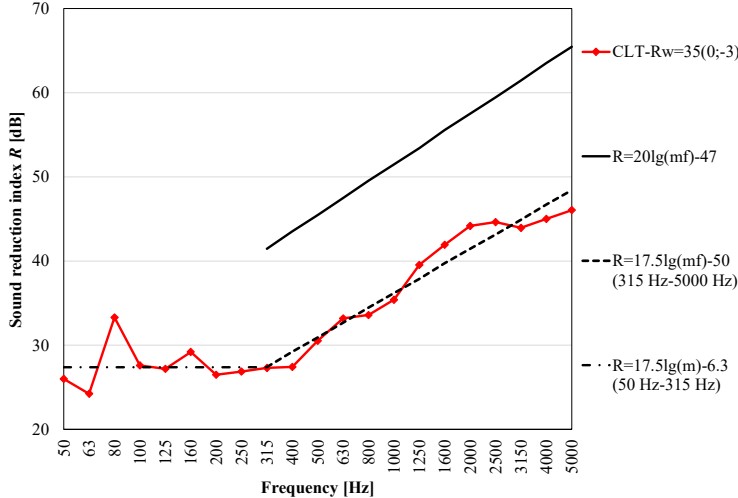

**Figure 10.** Measurements of sound reduction index for the 200 mm CLT floor and comparison with proposed mass law and $R = 20\lg(mf) - 47$.

1.  Starting from 315 Hz, there is an increasing trend with mass law $R = 17.5\lg(mf) - 50$, so between 50 Hz and 315 Hz, the equation $R = 17.5\lg(m') - 6.3$ can be used ($R^2 = 0.93$);
2.  The trend of the mass law (Equation (11)) is significantly different from the theoretical one [70].

Therefore, the classical mass law seems not to be suitable to characterize the acoustic insulation to the airborne noise of a CLT floor. The possible explanation could be related to the fact that the classical law of mass is obtained for thin panels, with a density greater than that of wood.

Figure 11 shows the results of a calculation performed, variating the modulus of elasticity and the loss factor, while input data, such as geometrical dimensions, were kept fix. The results of the simulations are then compared with the measured sound reduction index. Even by means of these software simulations, it is possible to observe an overestimation of the results, particularly in the medium-high frequencies. It seems, quite clearly, that a dedicated study for CLT is necessary, since none of the models manage to fairly calculate the requested characteristics.

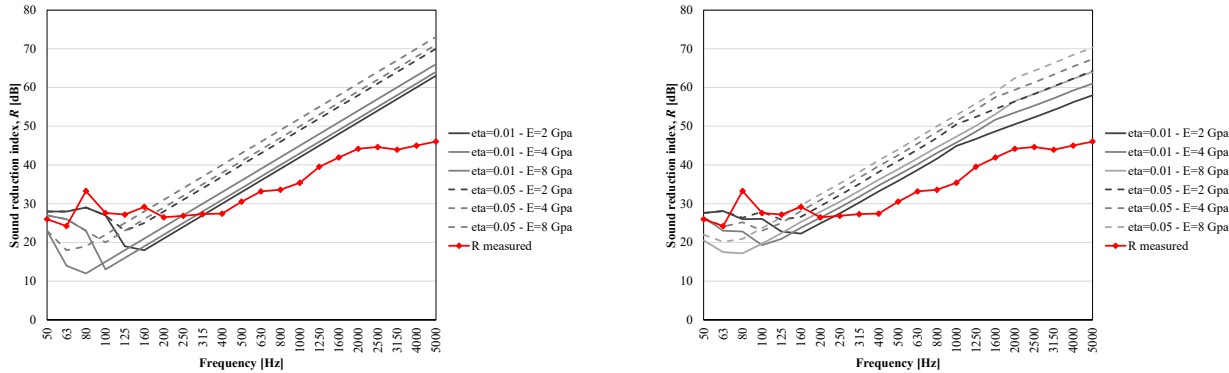

**Figure 11.** Comparison of measurement and simulation with two different commercial software.

Finally, the ISO 12354-2 [71] standard reports the following equation that correlates the sound reduction index, *R*, to the normalized impact sound pressure level, $L_n$:

$$R + L_n = 38 + 30\lg(f) \tag{11}$$

where both $R + L_n$ and the equation $38 + 30\lg(f)$ are reported in Figure 12.

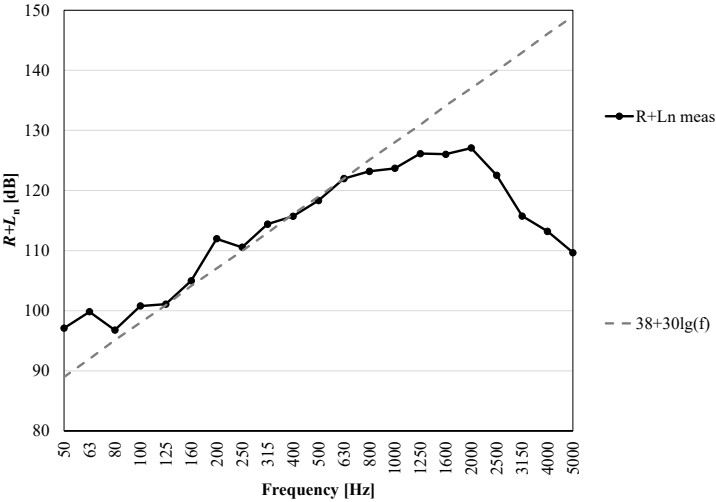

**Figure 12.** Measurements of normalized impact sound pressure level for the 200 mm CLT floor, comparison with $R + L_n = 38 + 30\lg(f)$ equation.

From the results in Figure 12, it can be seen that a poor correlation index ($R^2 = 0.72$) is obtained in the 100–3150 Hz frequency range, while a good correlation index ($R^2 = 0.95$) can be noted in the 100–2000 Hz frequency range.

### 3.2. Experimental Results: Vibration-Derived Parameters

A grid of accelerometer measurement positions is used to evaluate the vibration velocity of the CLT slab. Firstly, the frequency trends are plotted to investigate the velocity levels. From the velocity levels in Figure 13, it is possible to observe how those produced by the rubber ball, for frequencies below 125 Hz, are higher than those produced by the tapping machine and the dodecahedron. Conversely, the velocity level's trend of the standard tapping machine and of the rubber ball are partially overlapped, only in the range between 160–250 Hz. For frequencies below 80 Hz, however, the levels recorded are lower than those of the rubber ball, but higher than those of the dodecahedron. For frequencies above 250 Hz, the velocity levels of the standard tapping machine are the highest of the three sources presented here.

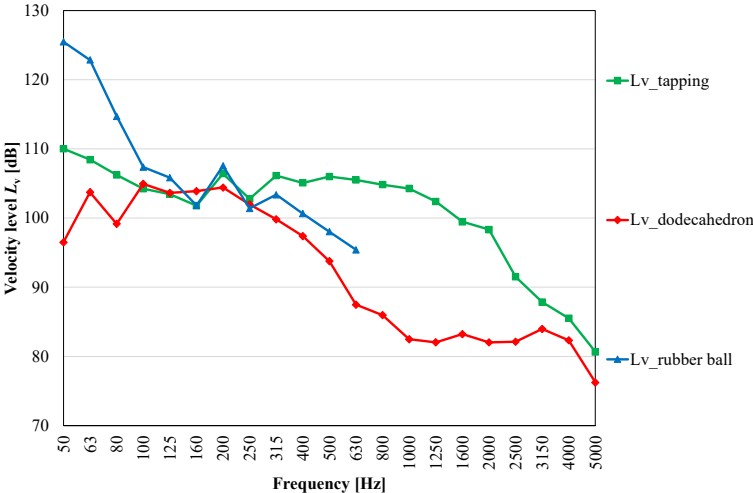

**Figure 13.** Measurements of velocity level: tapping machine (50–5000 Hz), dodecahedron (50–5000 Hz) and rubber ball (50–630 Hz).

Figure 14 shows the radiation indices of the CLT for each exciting source (tapping machine, airborne, noise and rubber ball). It can be clearly seen that, between 50 Hz and 80 Hz, the radiation indices of the two impactive sources are almost superimposed, while after 80 Hz, the value of the tapping machine is far higher, not only than the radiation index of the rubber ball, but also than that of the dodecahedron. From this analysis, it is evident how each sound source produces a different type of excitation. It is, therefore, important that all three sources are considered, when analyzing the behavior of a CLT building element.

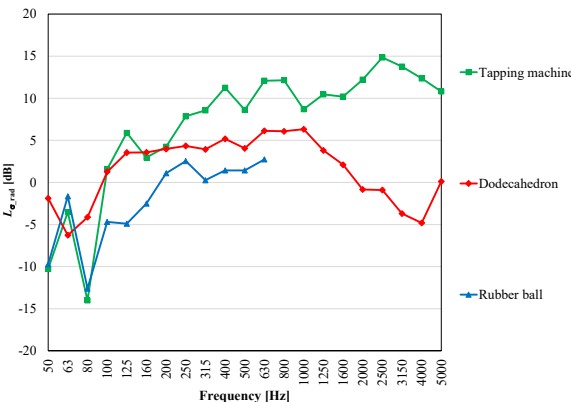

**Figure 14.** Measurements of radiation index: tapping machine (50–5000 Hz), dodecahedron (50–5000 Hz) and rubber ball (50–630 Hz).

Finally, by looking at the average of the nine measurements and at the variability of values for the nine measurements of the radiation indices, for each of the three sources (Figure 15), some considerations can follow.

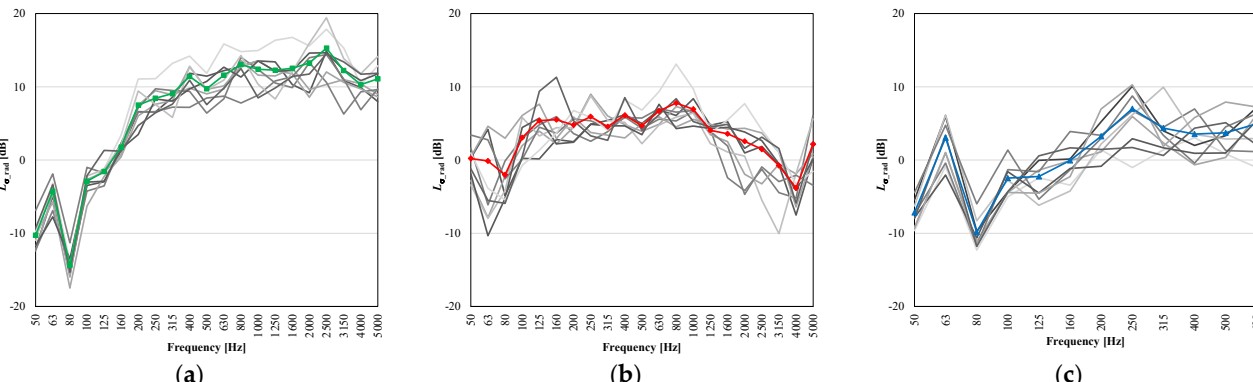

**Figure 15.** 9 points receiving measurement radiation indices and average respectively of: tapping machine (**a**), dodecahedron (**b**), rubber ball (**c**).

Firstly, looking at the tapping machine measurements, it is clear that, especially in the range between 50–80 Hz, all nine measurements are quite close to each other and to the average values. From 100 Hz to 200 Hz, they are rather scattered, then especially from 250 Hz onwards, the nine measurements seem quite different from each other, rarely overlapping the average curve.

Then, when looking at the measurements performed with the dodecahedron, it can be noted how the trends of each of the nine positions are quite dissimilar for almost all the frequencies. A more similar behavior of the nine curves, on the other hand, can be observed in the proximity of high frequencies (3150–5000 Hz).

Finally, by looking at the rubber ball trends, it is possible to note that (i) the trends are more similar to that of the averaged values at low frequencies (50–100 Hz) and that (ii) the differences between the nine positions, at high frequencies, are much more evident than at low frequencies.

Using the vibration data as a baseline, it was possible to derive and study the radiation index caused by each individual source at different receiver positions. This resulted in mapping, useful to better understand how the entire CLT floor surface behaves (Appendix A).

Looking at the maps produced by the impact of the tapping machine, some considerations can be made for the radiation index. The resonance peak of the normalized impact sound pressure level at 63 Hz is the same as the one of the corresponding radiation index. The negative peak at 80 Hz is also found in both $L_\text{n}$ and in $L_{\sigma_{rad}}$.

As in the case of the tapping machine, and also for the airborne noise, the peak at 63 Hz is clearly shown in Figure A2. Then, in accordance with the trend found in Figures 14 and 15, the minimum is reached at 80 Hz. Conversely to the tapping machine case, a fluctuation between negative and positive values of the radiation index can be seen. Then, if the overall average value is around −4 dB (dark green), a level of 3.0 dB is found in the centre-right part of the floor. Next, from 100 Hz to 2500 Hz, the surface maps always show a positive behavior. Then, some negative trends are found at 3150 Hz and 4000 Hz. Focusing on 3150 Hz, a negative trend of the radiation index is found in the centre-right part of the floor, while at 4000 Hz, the negative behavior of the radiation index characterizes the upper-right, together with centre-right area, and the low-centre area of the CLT flooring.

Finally, looking at the radiation index of the rubber ball (Figure A3), two down peaks characterize the impact source behavior on the CLT floors, at 50 and 80 Hz, in accordance with the trends of Figures 12 and 13. While the maximum level is found at 63 Hz, from 100 Hz onwards, an increasingly positive trend of the radiation index is always found.

For the impact noise produced by the tapping machine and the rubber ball, Equations (7) and (9) are proposed. These equations have been obtained by modifying those reported in

the ISO 12354-2 standard, minimizing the sum of the absolute value of the differences in the measured and theoretical levels. The results are shown in Figure 16.

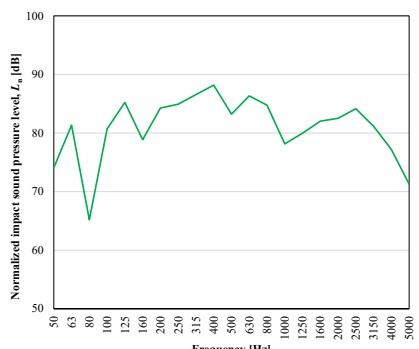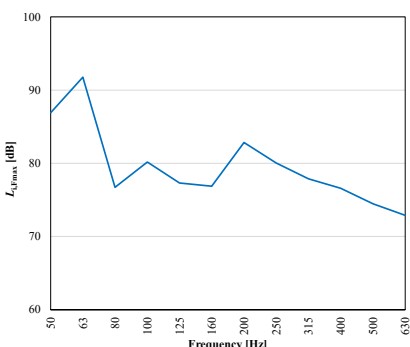

**Figure 16.** Simulated normalized sound reduction index (Equation (7)) and energy average maximum impact sound pressure level (Equation (9)).

## 4. Comparison between Measured and Calculated Values

It is now possible to use the sound radiation and velocity level, experimentally determined, together with the geometrical features, to predict the impact noise by means of the literature and proposed methods (equations presented in Sections 2.1 and 2.2), in order to compare them with the measured one.

In Figure 17, the normalized impact sound pressure level measured in the laboratory ($L_{n\_meas.}$), the expected level according to the reference curve ($L_{n\_ref.curve}$) [13], the level derived from the velocity levels ($L_{n\_Lv}$) according to Equation (5) and the expected level using the model in ISO 12354-2 ($L_{n\_12354-2}$), using Equation (6), and the expected level ($L_{n\_pr}$), using the proposed Equation (7) are shown.

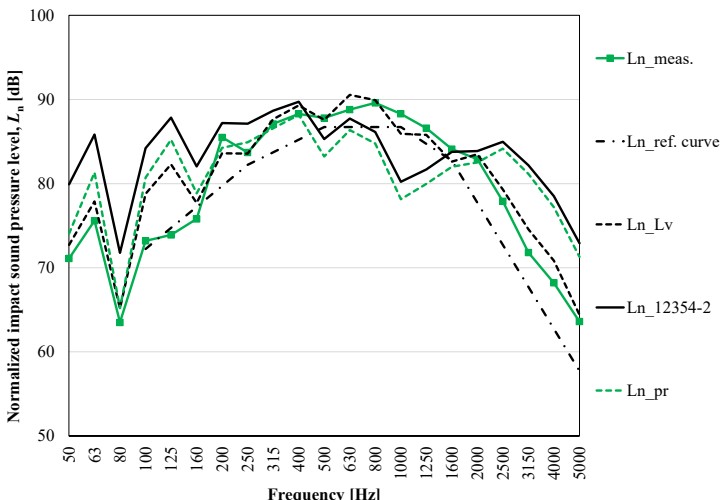

**Figure 17.** Measured and predicted normalized impact sound pressure level.

Table 4 shows the weighted normalized impact sound pressure levels for the measurements and models used.

**Table 4.** Weighted normalized impact sound pressure level, $L_{n,w}$.

| $L_{n,w\_meas}$ [dB] | $L_{n,w\_ref. curve}$ [dB] | $L_{n,w\_Lv}$ [dB] | $L_{n,w\_12354-2}$ [dB] | $L_{n,w\_pr}$ [dB] |
|---|---|---|---|---|
| 88 | 86 | 89 | 90 | 88 |

It can be noted that for all the models, the calculated values fall within a range of 2 dB. The result obtained using the velocity levels is more accurate, as it is derived from an actual

measurement made on the floor. It can be noted that the shape of the measured spectrum is similar to the reference curve [13], while the measured values are 2 dB higher than the reference curve.

Figure 18 shows the sound reduction index measured in the laboratory ($R_{\mathrm{meas}}$), according to the reference curve ($R_{\mathrm{ref.curve}}$), derived from the velocity levels and the levels measured in the transmitting room ($R_{\mathrm{Lv}}$), in which A ed L1 are the measured values, while L2 is evaluated according to Equation (4), and the predicted level using the model in ISO 12354-1 annex B ($R_{12354-1}$) and predicted with proposed equations $R = 17.5\lg(mf) - 50$, between 315 Hz and 5000 Hz and $R = 17.5\lg(m') - 6.3$, between 50 Hz and 315 Hz.

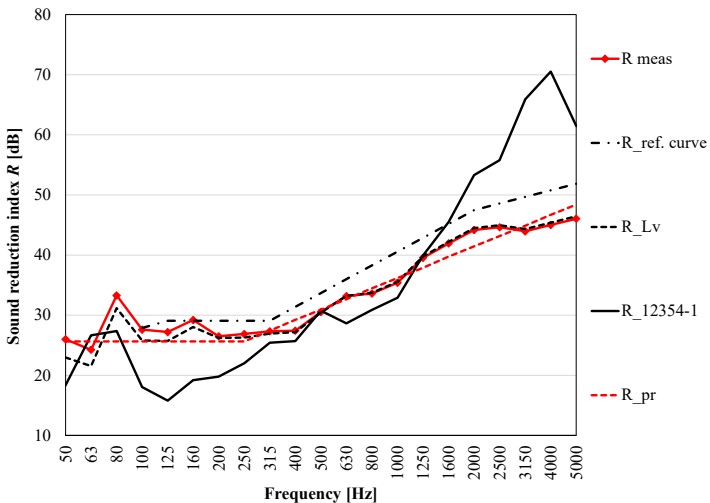

**Figure 18.** Measured and predicted sound reduction index.

Table 5 shows the weighted sound reduction indices for the measurements and models used. The prediction made by velocity level and sound pressure level measurements in the transmitting room is quite accurate, because actual measurements are used on the floor. The reference curve overestimates the performance by 4 dB, while the ISO 12354-1 model (as from the equations in paragraph 2.1) overestimates (above 1600 Hz) and underestimates (between 80 and 1000 Hz) the performance of the floor, significantly. The proposed equations lead to the same index $R_{\mathrm{w}}$ and spectrum adaptation terms $C$ and $C_{\mathrm{tr}}$.

**Table 5.** Weighted sound reduction indices, $R_{\mathrm{w}}$ ($C;C_{\mathrm{tr}}$ ).

| $R_{\mathrm{w}}$ [dB] | $R_{\mathrm{w\_ref.\ Curve}}$ [dB] | $R_{\mathrm{w\_Lv}}$ [dB] | $R_{\mathrm{w\_12354-1}}$ [dB] | $R_{\mathrm{w\_pr}}$ [dB] |
|---|---|---|---|---|
| 35(0; −3) | 39(−1; −4) | 35(0; −3) | 32(−1; −4) | 35(0; −3) |

Figure 19 shows the maximum sound pressure level measured in the laboratory ($L_{\mathrm{i,\ Fmax\_meas}}$), the level derived from the velocity levels ($L_{\mathrm{i,Fmax\_Lv}}$) according to Equation (4) and the predicted level using the model in ISO 12354-2 ($L_{\mathrm{i,Fmax\_12354-2}}$), using Equation (8) and the expected level ($L_{\mathrm{i,Fmax\_pr}}$), using the proposed Equation (7).

Table 6 shows the A-weighted sound pressure levels for the measurements and models used. The results show that looking at the weighted indices, the prediction made with the model derived from ISO 12354-2 gives a better value (+1 dB difference) than the model based on measured velocity levels (−2 dB difference). This results from the fact that the A-weighting gives an increasing weighting, as the frequency increases from 50 Hz to 630 Hz. When analyzing the frequency trend, it can be seen that the model derived from the measured velocity values is significantly more accurate. The average absolute value of the differences is 1.7 dB, using $L_{\mathrm{v}}$ and 3.6 dB, using the model derived from 12354-2. The proposed equations lead to the same weighted index $L_{\mathrm{i,\ AFmax}}$.

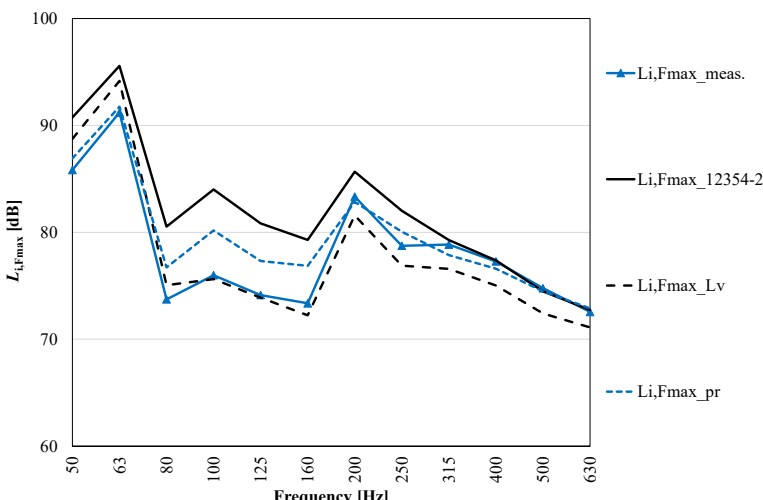

**Figure 19.** Measured and predicted energy average maximum impact sound pressure level with rubber ball.

**Table 6.** Weighted sound pressure levels.

| $L_{iA,Fmax\_meas}$ [dB(A)] | $L_{iA,Fmax\_Lv}$ [dB(A)] | $L_{iA,Fmax\_12354-2}$ [dB(A)] | $L_{iA,Fmax\_pr}$ [dB(A)] |
|---|---|---|---|
| 80 | 78 | 81 | 80 |

## 5. Conclusions

This work presents the experimental results of acoustic measurements, carried out on a CLT floor, consisting of five layers of 200 mm thickness, according to the ISO 10140 series standards.

1.  Firstly, three different types of sources were used in the laboratory of the Free University of Bolzano: tapping machine, dodecahedron and rubber ball. Furthermore, vibration measurements were carried out for each source. The results make it possible to derive the impact sound pressure level obtained from the tapping machine and the rubber ball, the sound reduction index obtained with the dodecahedron, velocity levels and consequent radiation index for each of the sources. For all three sources, the use of velocity level measurements to derive the acoustic parameters has led to excellent results, in comparison to the experimental data. This is not a predictive method and could not be fully verified in in-situ measurements, due to lateral transmissions, which are not trivial, to be determined., As regards the measurement of the radiation index, it can be noted that at the frequency of 63 Hz, the higher value can be found in correspondence with the resonance frequency of the floor for each type of sound source. The same maximum can be found in the sound pressure measurements. Furthermore, the radiation indices of the two impact sources (tapping machine and rubber ball), at low frequency (50–80 Hz), provided results significantly similar, while after 80 Hz, the values of the tapping machine far exceed not only the radiation index of the rubber ball, but also that of the dodecahedron.

2.  Then, it was observed that the predicted sound reduction index, evaluated with the mass law $R = 20\lg(mf) - 47$, was not suitable for the CLT floor examined, and a new proposed law was introduced for CLT floors. Furthermore, a correlation between the normalized impact pressure level and the impact sound pressure level of the rubber ball was detected.

3.  Finally, for all three types of sources, the use of velocity level measurements to derive the acoustic parameters has led to good results, in comparison to the experimental data. This could not be fully verified in in-situ measurements due to lateral transmissions, which are not trivial, to be determined.

As for the impact sound pressure level, good correlations (within 2 dB) can be seen, both with the use of the reference curve, and by using the formula provided by the ISO 12354-2 standard, using, in any case, the measured radiation index and the structural reverberation time. As for the sound reduction index, the standard model leads to an underestimation at medium-low frequencies and to a considerable overestimation at medium-high frequencies. From the results obtained, it was possible to see how traditional analytical models do not provide good approximations of the measurements. Good correlations are obtained using the proposed equations; however, these correlations can be used with floors having characteristics similar to those used in laboratory tests.

A future development of this work will concern the study of the CLT floor with coverings consisting of acoustic mats, with a concrete screed and the influence of the physical characteristics of the materials, with respect to the acoustic performance.

**Author Contributions:** Conceptualization, N.G.; software, N.G.; project administration, N.G.; validation, N.G. and A.M.; methodology, N.G. and A.M.; formal analysis, N.G. and A.M.; investigation, N.G. and A.M.; data curation, N.G. and A.M.; writing—original draft preparation, N.G. and A.M.; writing—review and editing, N.G. and A.M.; visualization, A.M.; supervision, A.G.; funding acquisition, N.G. and A.M. All authors have read and agreed to the published version of the manuscript.

**Funding:** This research received no external funding.

**Institutional Review Board Statement:** Not applicable.

**Informed Consent Statement:** Not applicable.

**Acknowledgments:** This work was financed by the European Interreg BIGWOOD project, ITAT 1081 CUP: I54I18000300006.

**Conflicts of Interest:** The authors declare no conflict of interest.

## Appendix A

*Appendix A.1. Tapping Machine Radiation Index Mapping*

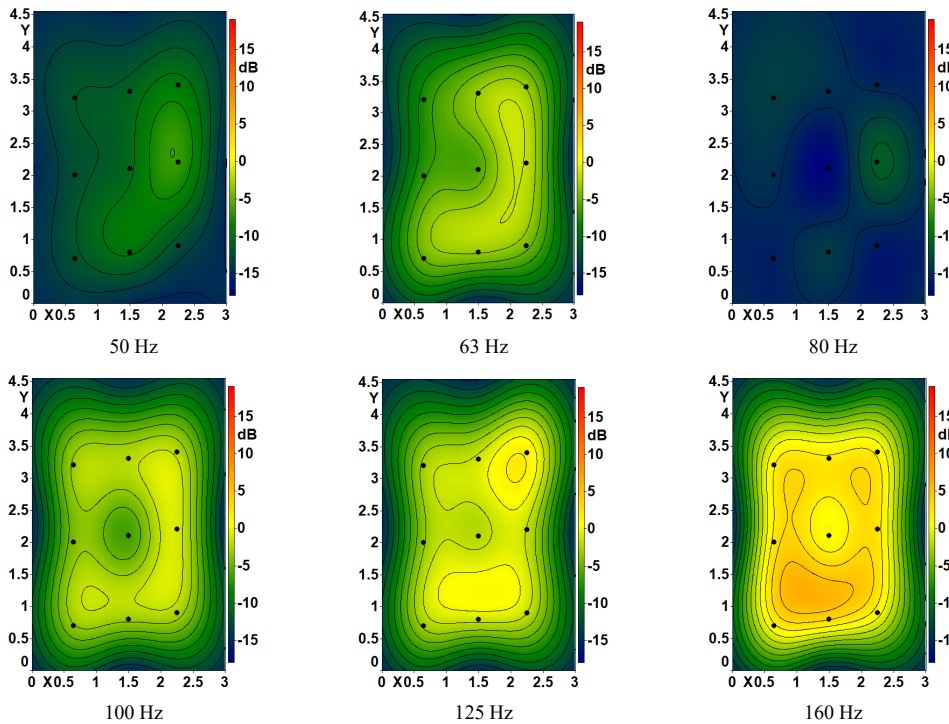

**Figure A1.** *Cont.*

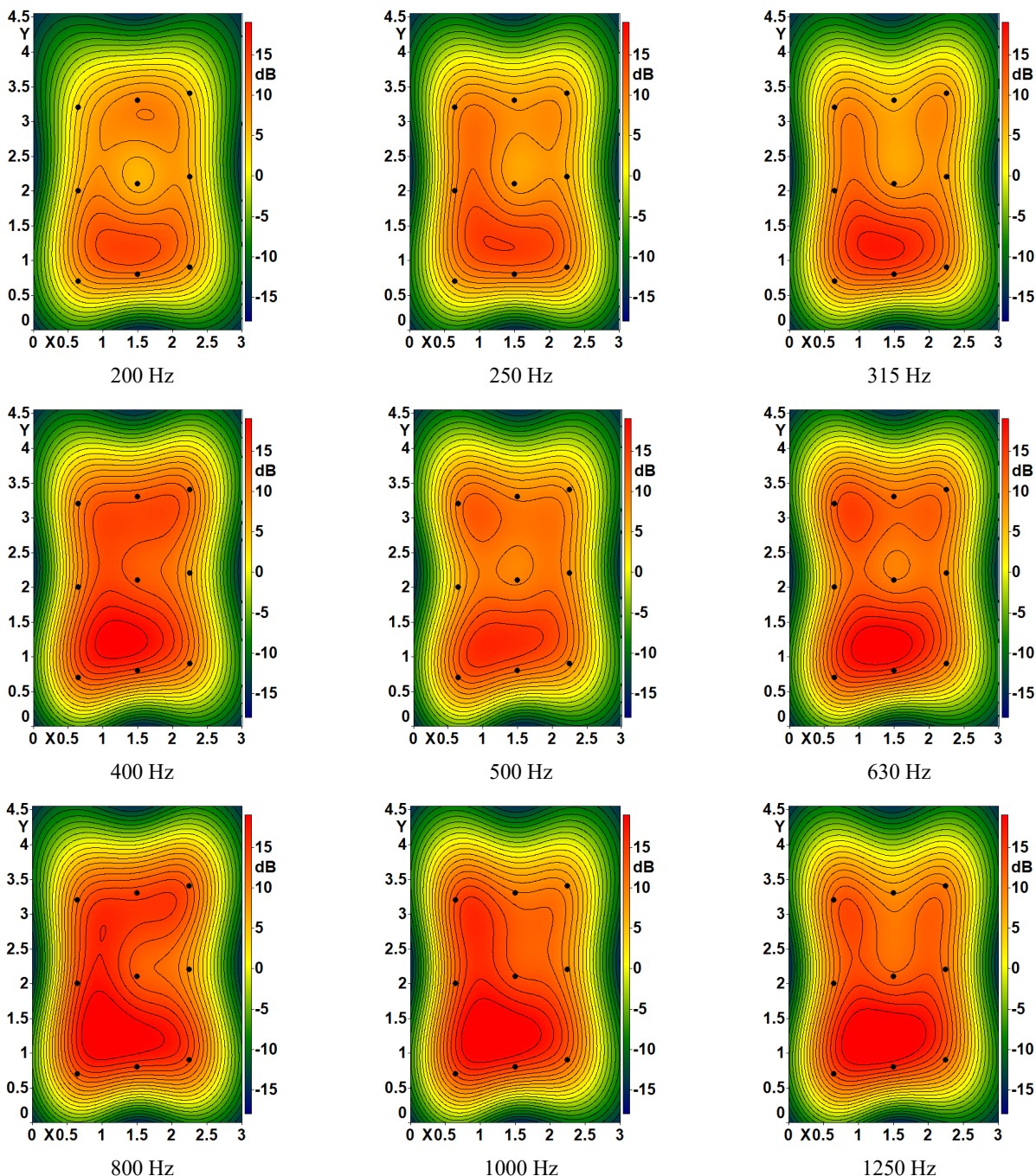

**Figure A1.** *Cont.*

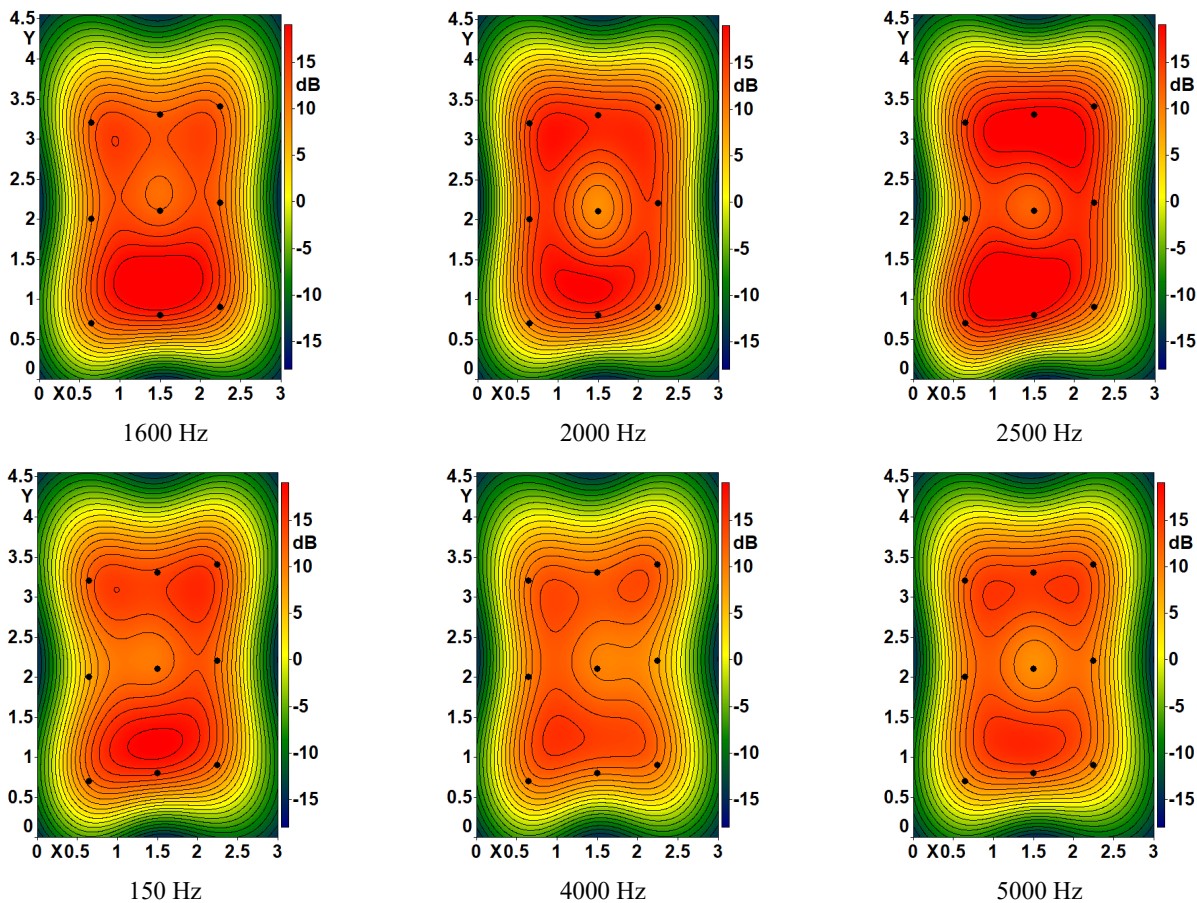

**Figure A1.** Tapping machine radiation index maps. Average between the three source positions.

*Appendix A.2. Airborne Noise Source Radiation Index Mapping*

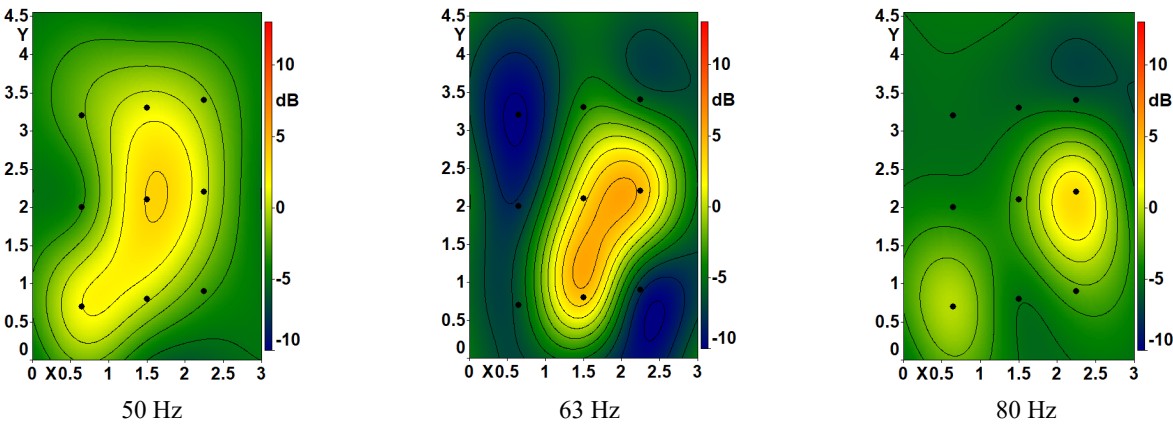

**Figure A2.** *Cont.*

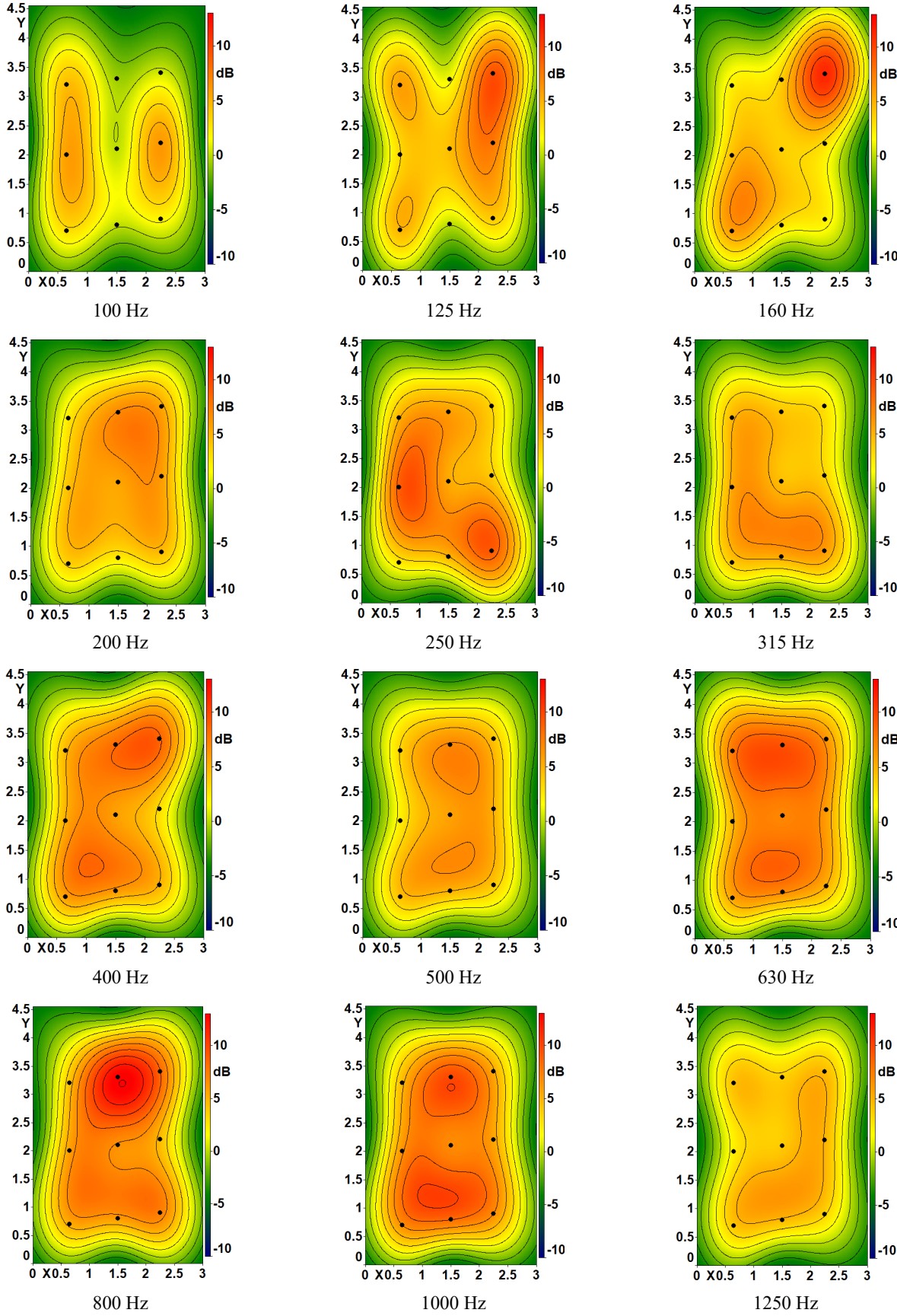

**Figure A2.** *Cont.*

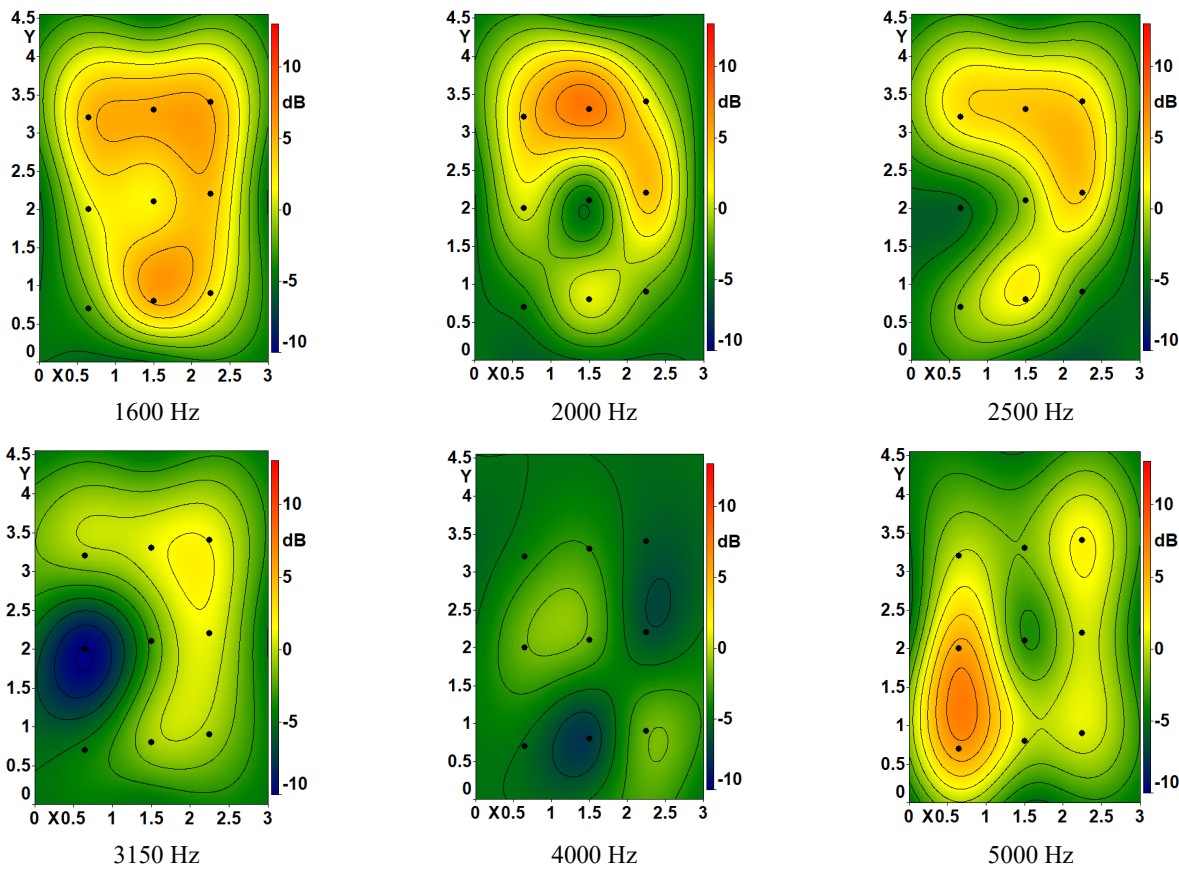

**Figure A2.** Air-borne source radiation index maps. Average between the three source positions.

*Appendix A.3. Rubber Ball Radiation Efficiency Mapping*

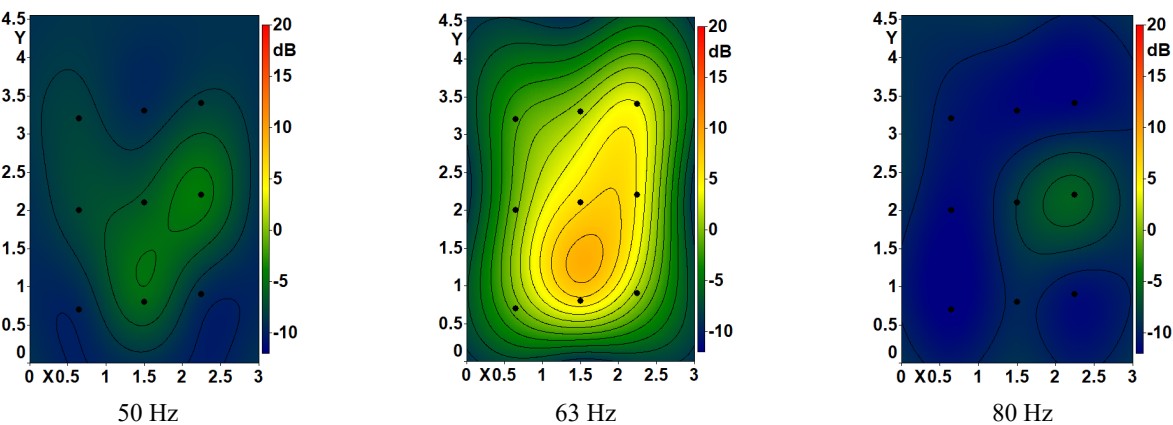

**Figure A3.** *Cont*.

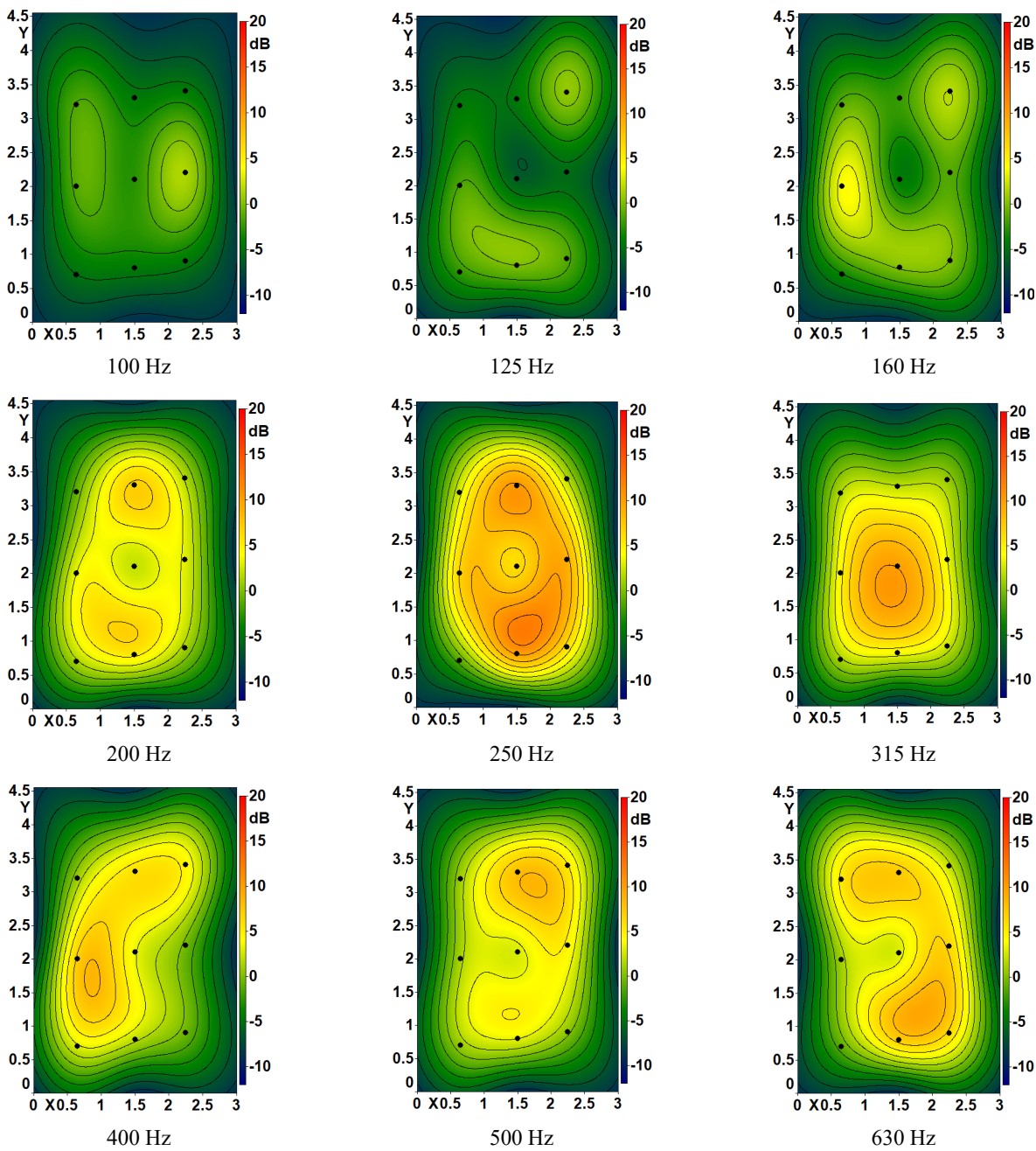

**Figure A3.** Rubber ball radiation index maps. Average between the three source positions.

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
