# Peer review of "Cross-Laminated Timber Floor: Analysis of the Acoustic Properties and Radiation Efficiency"

_applsci, doi:10.3390/app12073233_

Round 1

Reviewer 1 Report

Section 1 must be improved. Authors should emphasize contribution and novelty, the introduction needs to clarify the motivation, challenges, contribution, objectives, and significance/implication.  At the end of the section, add an outline of the rest of the paper, in this way the reader will be introduced to the content of the following sections.

Section 2 must be improved. A detailed description of the rooms in which the measures were carried out is missing. You need a detailed description of the floor, walls and ceiling. This is to make it clear how the noise is being transmitted. Describe in detail the equipment used to make the measurements (airborne noise source, tapping machine, and rubber ball). Extract this data from the datasheet of the instrumentation manufacturer. To make reading the specifications of the instruments more immediate, you can insert them in a table, listing the instruments used and the specific characteristics for each. Also you need to check the format of the figures, tables and equations.

Section 3 must be improved. A section for discussion of the results obtained is missing. You should explain in detail, based on your assumptions, what the results were obtained and how those results allow you to draw conclusions.

Section 4 must be improved. I have not found an adequate discussion of the analytical model that the authors claimed to propose in this study (See Introduction).

Section 5 must be improved. Paragraphs are missing where the possible practical applications of the results of this study are reported. What these results can serve the people, it is necessary to insert possible uses of this study that justify their publication. They also lack the possible future goals of this work. Do the authors plan to continue their research on this topic?

References: I noticed that an author is cited with a percentage higher than 23%, it is suggested to carry out a check.

53-59) This section is superfluous and does not add any content to the paper. It is suggested to remove it, as well as Figure 1.

98-99) “partially incomplete” Remove this statement it's your opinion.

112-117) Check the format of this section

118) Figure 2 needs improvement, add a label to the two figures (a) and (b). Add these labels in the caption. Describe in the caption what are the differences that can be appreciated in the two figures.

124) Check the format of table 1, is not the mdpi format. I have seen that you often use this format, so I will not repeat this advice again, it also applies to the other occurrences. The table header must start with a capital letter.

142) Figure 3 needs improvement, add a label to the three figures (a) , (b) and (c). Add these labels in the caption.

145-159) Specify how the two rooms are arranged (transmitting room, receiving room)

161) Figure 4 needs improvement, add a label to the figures. Add these labels in the caption. Remove left, center, right.

170) Figure 5 needs improvement, add a label to the figures. Add these labels in the caption.

177-180) Check the format of the equations, the number of the equation goes to the right. I have seen that you often use this format, so I will not repeat this advice again, it also applies to the other occurrences.

255) Figure 7 needs improvement, add a label to the figures. Add these labels in the caption.

319) Figure 10 needs improvement. Too many curves, it is not possible to distinguish them well.

Author Response

Dear reviewer,

Thank you very much for the helpful comments on the manuscript. They have been all considered in the new version of the article. You can find a point-by-point reply to all the points hereafter.

  1. Section 1 must be improved. Authors should emphasize contribution and novelty, the introduction needs to clarify the motivation, challenges, contribution, objectives, and significance/implication.  At the end of the section, add an outline of the rest of the paper, in this way the reader will be introduced to the content of the following sections.

Thank you for your comment! Section 1 has been improved, furthermore, an outline has been added at the end of the introduction as suggested.

  1. Section 2 must be improved. A detailed description of the rooms in which the measures were carried out is missing. You need a detailed description of the floor, walls and ceiling. This is to make it clear how the noise is being transmitted. Describe in detail the equipment used to make the measurements (airborne noise source, tapping machine, and rubber ball). Extract this data from the datasheet of the instrumentation manufacturer. To make reading the specifications of the instruments more immediate, you can insert them in a table, listing the instruments used and the specific characteristics for each. Also, you need to check the format of the figures, tables and equations.

Thank you for your useful suggestion. Section 2 has been improved, also equipment details have been added. The format of the figure, table and equations has been correctly modified all over the paper according to the template as suggested.

  1. Section 3 must be improved. A section for discussion of the results obtained is missing. You should explain in detail, based on your assumptions, what the results were obtained and how those results allow you to draw conclusions.

Thank you for the observation. Some further comments in the discussion and results section were added, with particular attention both to the sound pressure levels and vibrational derived parameters section.

  1. Section 4 must be improved. I have not found an adequate discussion of the analytical model that the authors claimed to propose in this study (See Introduction).

Thank you for your observation! A discussion on the analytical model was further detailed.

  1. Section 5 must be improved. Paragraphs are missing where the possible practical applications of the results of this study are reported. What these results can serve the people, it is necessary to insert possible uses of this study that justify their publication. They also lack the possible future goals of this work. Do the authors plan to continue their research on this topic?

Thank you for the comment. A short paragraph regarding future research and development was added.

  1. References: I noticed that an author is cited with a percentage higher than 23%, it is suggested to carry out a check.

Thank you for your observation, some references of that Author were deleted were strictly not necessary for the topics addressed.

  1. 53-59) This section is superfluous and does not add any content to the paper. It is suggested to remove it, as well as Figure 1.

Accordingly on what suggested by the reviewer, Figure 1 has been removed.

  1. 98-99) “partially incomplete” Remove this statement it's your opinion.

Thank you for the feedback, the statement “partially incomplete” has been removed.

  1. 112-117) Check the format of this section

Thank you for the feedback, the format has been corrected.

  1. 118) Figure 2 needs improvement, add a label to the two figures (a) and (b). Add these labels in the caption. Describe in the caption what are the differences that can be appreciated between the two figures.

Thank you for your comment, Figure 2 label has been improved and modified as suggested.

  1. 124) Check the format of table 1, is not the mdpi format. I have seen that you often use this format, so I will not repeat this advice again, it also applies to the other occurrences. The table header must start with a capital letter.

Thank you for the feedback. The format of table 1 has been modified according to the template you suggested.

  1. 142) Figure 3 needs improvement, add a label to the three figures (a) , (b) and (c). Add these labels in the caption.

Thank you for your comment, Figure 3 label has been improved and modified as suggested.

  1. 145-159) Specify how the two rooms are arranged (transmitting room, receiving room)

Thank you for your comment, A description of the details on how the two rooms were settled was added.

  1. 161) Figure 4 needs improvement, add a label to the figures. Add these labels in the caption. Remove left, centre, right.

Thank you for your comment, Figure 4 label has been improved and modified as suggested.

  1. 170) Figure 5 needs improvement, add a label to the figures. Add these labels in the caption.

Thank you for your comment, Figure 5 label has been improved and modified as suggested.

  1. 177-180) Check the format of the equations, the number of the equation goes to the right. I have seen that you often use this format, so I will not repeat this advice again, it also applies to the other occurrences.

Thank you for the comment. The format of the equations has been modified according to the template as suggested.

  1. 255) Figure 7 needs improvement, add a label to the figures. Add these labels in the caption.

Thank you for your comment, Figure 7 label has been improved and modified as suggested.

  1. 319) Figure 10 needs improvement. Too many curves, it is not possible to distinguish them well.

Thank you for your comment. The number of the curves in Figure 10 has been reduced in order to have a better look at the results.

Reviewer 2 Report

A. Hosseinkhani, D. Younesian, A. O. Krushynska, M. Ranjbar, F. Scarpa, “Full-gradient optimization of the vibro-acoustic performance of (non-)auxetic sandwich panels”, Transport in Porous Media (2021), https://doi.org/10.1007/s11242-021-01693-0

M. S. Mazloomi, M. Ranjbar, “Hybrid Design Optimization of Sandwich Panels with Gradient Shape Anti-Tetrachiral Auxetic Core for Vibroacoustic Applications,” Transport in Porous Media, (2021), https://doi.org/10.1007/s11242-021-01646-7

E. Panahi, A. Hosseinkhani, M. F. Khansanami, M. Ranjbar, D. Younesian, “Novel Cross Shape Phononic Crystals with Broadband Vibration Wave Attenuation Characteristics: Design, Modeling and Testing”, Thin-Walled Structures Journal 163 (2021) 107665

A. Hosseinkhani, D. Younesian, M. Ranjbar, F. Scarpa, “Enhancement of the Vibro-acoustic Performance of Anti-tetra-chiral Auxetic Sandwich Panels Using Topologically Optimized Local Resonators”, Applied Acoustics 177 (2021) 10793

Please improve the introduction by referring to the provided articles.

Author Response

Dear reviewer,

Thank you very much for the helpful comments to the manuscript. They have been all considered in the new version of the article. You can find a point by point reply to all the points hereafter.

  1. Hosseinkhani, D. Younesian, A. O. Krushynska, M. Ranjbar, F. Scarpa, “Full-gradient optimization of the vibro-acoustic performance of (non-)auxetic sandwich panels”, Transport in Porous Media (2021), https://doi.org/10.1007/s11242-021-01693-0
  2. S. Mazloomi, M. Ranjbar, “Hybrid Design Optimization of Sandwich Panels with Gradient Shape Anti-Tetrachiral Auxetic Core for Vibroacoustic Applications,” Transport in Porous Media, (2021), https://doi.org/10.1007/s11242-021-01646-7
  3. Panahi, A. Hosseinkhani, M. F. Khansanami, M. Ranjbar, D. Younesian, “Novel Cross Shape Phononic Crystals with Broadband Vibration Wave Attenuation Characteristics: Design, Modeling and Testing”, Thin-Walled Structures Journal 163 (2021) 107665

Hosseinkhani, D. Younesian, M. Ranjbar, F. Scarpa, “Enhancement of the Vibro-acoustic Performance of Anti-tetra-chiral Auxetic Sandwich Panels Using Topologically Optimized Local Resonators”, Applied Acoustics 177 (2021) 10793

Please improve the introduction by referring to the provided articles.

Thank you for your comment. The following papers have been added:

  1. Hosseinkhani, D. Younesian, A. O. Krushynska, M. Ranjbar, F. Scarpa, “Full-gradient optimization of the vibro-acoustic performance of (non-)auxetic sandwich panels”, Transport in Porous Media (2021); https://doi.org/10.1007/s11242-021-01693-0
  2. Hosseinkhani, D. Younesian, M. Ranjbar, F. Scarpa, “Enhancement of the Vibro-acoustic Performance of Anti-tetra-chiral Auxetic Sandwich Panels Using Topologically Optimized Local Resonators”, Applied Acoustics 177 (2021) 10793; https://doi.org/10.1016/j.apacoust.2021.107930

Reviewer 3 Report

For a clear explanation of the research results, some detailed descriptions would like to be helpful in the laboratory where the experiments were conducted and the installation condition of the specimen. 

In Figures 4 and 5, the dimension of each specimen and experimental setup was different for airborne sound insulation and floor impact sound insulation. The authors have to explain the reason for the different dimensions of the specimens.

Page 9, Line 306: The authors find out that the classical mas laws are not suitable to the airborne sound insulation of the CLT floor. If discussions on the reason why the classical mass law was not suitable for the CLT floor, are added it would be helpful to understand the results of this paper. 

Page 12, Line 372~375: If the authors have an impact force or vibrational characteristics of real impact sources, it is needed to compare to the characterises of the tapping or rubber ball. 

For the Figures in Appendix A, just measurement positions were presented in the figures. For an easy understanding of the experimental results, the source position shall be presented in the figures.

Editorial issues

  • Page 7, Line 243: Some wordings not related to the topic of this paper is in the manuscripts.
  • Please check Figure 12 and Figure 14, Both Figures looks like the same one.
  • Figure 13 is presented on Page 12, but the same figures are also on page 13.

Author Response

Dear reviewer,

Thank you very much for the helpful comments to the manuscript. They have been all considered in the new version of the article. You can find a point-by-point reply to all the points hereafter.

  1. For a clear explanation of the research results, some detailed descriptions would like to be helpful in the laboratory where the experiments were conducted and the installation condition of the specimen. 

Thank you for your comment. An additional explanation regarding the construction of the chambers and the constraint conditions of the specimen has been added

  1. In Figures 4 and 5, the dimension of each specimen and experimental setup was different for airborne sound insulation and floor impact sound insulation. The authors have to explain the reason for the different dimensions of the specimens.

Thank you for your feedback. The CLT specimen is the same for all the measurements made (dimensions 3000 mm x 4155 mm). What changes is the plan surface of the transmitting room and the receiving room (the receiving room is larger than the transmitting one). A short explanatory sentence was added to the text.

  1. Page 9, Line 306: The authors find out that the classical mas laws are not suitable to the airborne sound insulation of the CLT floor. If discussions on the reason why the classical mass law was not suitable for the CLT floor, are added it would be helpful to understand the results of this paper. 

Thank you for your comment. We added an explanatory sentence. Indeed, an explanation for the not suitability of the mass law could be related to the fact that the classical law of mass is obtained for thin panels with a density greater than that of wood.

  1. Page 12, Line 372~375: If the authors have an impact force or vibrational characteristics of real impact sources, it is needed to compare to the characterises of the tapping or rubber ball.

Thank you for the feedback. In this work the real impact force on the floor has not been determined.

  1. For the Figures in Appendix A, just measurement positions were presented in the figures. For an easy understanding of the experimental results, the source position shall be presented in the figures.

Thank you for your comment. Since the maps have been made by averaging the levels of the 3 source positions, the positions of the sources have not been inserted in order not to weigh down the reading of the maps themselves.

  1. Editorial issues

Page 7, Line 243: Some wordings not related to the topic of this paper is in the manuscripts.

This part has been deleted.

Please check Figure 12 and Figure 14, Both Figures looks like the same one.

The figure has been modified.

  1. Figure 13 is presented on Page 12, but the same figures are also on page 13.

Reviewer 4 Report

It gives the impression that the specific dimensions of the specimen affect its own resonant frequency as a plate and that its own nodes are produced which affect the results of the experiment.

There is a lack of photographic and schematic information on the specific configuration of the experiment.

The conclusions are poor compared to the extensive prior analysis of the results.

A comparison with an isotropic material already known to better contrast the results is missing.

The results are likely to vary over time as the material changes its moisture or acquires deferred deformation.

Author Response

Dear reviewer,

Thank you very much for the helpful comments on the manuscript. They have been all considered in the new version of the article. You can find a point by point reply to all the points hereafter.

  1. It gives the impression that the specific dimensions of the specimen affect its own resonant frequency as a plate and that its own nodes are produced which affect the results of the experiment.

Thank you for your observation. The CLT specimen has a resonant frequency characteristic of its size, this frequency is equal to 63 Hz as mentioned also in the paper. The resonance frequency is always present in building structures tested in the laboratory because the dimensions of the specimen are relatively small (for floors including the surface between 10 m2 and 20 m2).

  1. There is a lack of photographic and schematic information on the specific configuration of the experiment.

Thank you for your feedback. A photo of the lab and CLT floor has been added as suggested.

  1. The conclusions are poor compared to the extensive prior analysis of the results.

Thank you for your comment. Some further comments were added in the conclusion section as suggested.

  1. A comparison with an isotropic material already known to better contrast the results is missing.

Thank you for your feedback! A figure has been added with a comparison between the wooden floor and the concrete isotropic floor to better understand the difference in behavior at high frequencies between the two floors.

  1. The results are likely to vary over time as the material changes its moisture or acquires deferred deformation.

Thank you for your comments. The wood of the floor is seasoned in such a way as to minimize the variability of the measures; to clarify these details to the readers a sentence was added at the beginning of Section 2.

Round 2

Reviewer 1 Report

The authors addressed all the reviewer's comments with sufficient attention and modified the paper with the suggestions provided. The new version of the paper has improved. Authors now need to focus on the paper format which needs to be revised significantly.

Minor revision

Authors need to improve the format of tables and figures. For example, figures 2 and 3 require an intervention, in particular figure 3 is very confusing, the sub-figures overlap. The numbers of many figures are totally wrong, passing from 3 to 43. The tables are not correctly positioned and some have the headings in black. All paper formatting needs to be revised.

Author Response

Thanks for the helpful feedback on the need to comply with the template. We really appreciate your comments.

According on what suggested, we modified the caption of Tables 4 and 5:

  • The character of the caption of Table 4 is from 11 to 10 as required by the template.
  • For Table 5 the font has been changed to Palatino Linotype.
  • We also checked that for each Table only the Top and the Bottom border of the Table were with a line width of 1pt, while the second horizontal line was settled on 0.5 pt as used in the Template.

To the other request, we can not see any other problem according to the template. The order of the tables according to the last version uploaded is sequential, all the captions have the bold layout only when referring to "Table/Figure" and not in the caption description. We would suggest to the Reviewer to check the file with the review mode activated with only simple markup. In this way, it should be simpler to check the compliance with the MDPI Template.
Anyway, we also attached a clean version of the paper in the zip file.

Thank you so much for all your useful feedback! We hope to have solved everything as requested!

Reviewer 3 Report

Thank you for the kind and detailed response to the review comments.

Also, appreciate the developed manuscript reflecting the review. 

Author Response

The authors thank the reviewer for their helpful comments and feedback.